# Psychometric Properties of the SV-RES60 Resilience Scale in a Sample of Chilean Elementary School Teachers

**DOI:** 10.3390/bs13090781

**Published:** 2023-09-20

**Authors:** Sonia Salvo-Garrido, Karina Polanco-Levicán, Sergio Dominguez-Lara, Manuel Mieres-Chacaltana, José Luis Gálvez-Nieto

**Affiliations:** 1Departamento de Matemática y Estadística, Universidad de La Frontera, Temuco 4780000, Chile; 2Programa de Doctorado en Ciencias Sociales, Universidad de La Frontera, Temuco 4780000, Chile; karinapolanco.l@gmail.com; 3South American Center for Education and Research in Public Health, Universidad Privada Norbert Wiener, Lima 15108, Peru; sergio.dominguez@uwiener.edu.pe; 4Departamento de Diversidad y Educación Intercultural, Universidad Católica de Temuco, Temuco 4780000, Chile; mieres@uct.cl; 5Departamento de Trabajo Social, Universidad de La Frontera, Temuco 4780000, Chile; jose.galvez@ufrontera.cl

**Keywords:** resilience, teachers, primary education, psychometric properties, ESEM

## Abstract

The concept of resilience, identified as a crucial variable due to its association with several beneficial outcomes in adulthood, is of particular interest in the teaching field. Specifically, teachers work in a demanding, challenging, and stressful context that requires a remarkable ability to adapt; therefore, resilience is important in the field of teaching and training, as it plays a fundamental role in children’s cognitive, social, and emotional development. This study sought to analyze the psychometric properties of the SV-RES60 Resilience Scale in a sample of Chilean elementary school teachers from first to eighth grade (N = 1406; mean age = 41.4; SD = 10.8). ESEM and bifactor ESEM analyses were performed to evaluate its factor structure, internal consistency, and reliability. The results supported a bifactor structure in which resilience was represented by one general latent factor and twelve specific factors (RMSEA = 0.032; 90%CI [0.030, 0.033]; SRMR = 0.012; CFI = 0.986; TLI = 0.977). A predominance of the unidimensional components of the SV-RES60 (general factor, ECV = 0.812; ωh = 0.975) and a high reliability (α = 0.981; ω of the general factor = 0.991) were observed. In conclusion, the SV-RES60 Resilience Scale is a suitable instrument for measuring the general factor of resilience in the investigated teaching environment. Future studies could contribute towards evidence of a reduced scale and transcultural validation to conduct comparative studies.

## 1. Introduction

Elementary teachers fulfill a fundamental role in their students’ cognitive, social, and emotional development [1]. Teaching is an emotional, physical, and intellectually challenging job [2], which is why resilience is a fundamental non-academic skill for teachers who work with children [3,4].

Teachers’ resilience plays a relevant role in their performance in the classroom and in their own training [5]. This is because environments change and demand constant adaptation to new demands of the context [6,7,8,9]. Teachers view teaching as a job that continues into their free time, with a high workload and high stress levels that affect their personal and professional lives; hence, resilience helps them to achieve their goals [10]. Moreover, it enables them to face difficulties using different strategies so that their students can learn [11]. According to Sharifian et al. [12], elementary teachers with high resilience cope more effectively with the complex situations they experience, resulting in less trauma and burnout.

Resilience is understood as a process that allows a person to face difficult situations [13] and is able to begin a personal growth project through the ability to reorganize while maintaining their inner balance despite external situations [14]. Saavedra and Villalta [15] suggest that resilience manifests in the resilient response that is derived from the person’s view of the problem, view of themselves, and the underlying conditions related to the person’s belief system. Therefore, according to these authors, resilience is built over time, giving the self a sense of continuity while transforming it in its interactions with the environment. In this reorganization process, the person benefits from interactions between different protective factors at the personal, school, social, and community levels, enabling new competencies to emerge to reach future goals [14,16]. Therefore, teachers’ resilience is an individual, relational, and collective process that occurs over time. Consequently, the characteristics present in the school context can trigger certain risks or difficulties that can be conceived as hindering or enhancing the processes of resilience [9,17,18,19,20]. For example, Rancher and Moreland [1] report that teachers who have felt a greater number of chronic stress factors of adverse experiences in childhood, and who experience stress at work, exhibit lower resilience levels.

Resilience is related to the quality of the working conditions [21], to the opportunities for professional development [6], and to relationships, support, and trust with colleagues [22,23]. In the same vein, resilience increases with school administrations that support teachers and with work resources, such as feedback, autonomy, flexibility, solidarity, and collective responsibility [9,24,25], whereas monitoring work tasks and behaviors and work overload decrease the perception of resilience [26]. In addition, resilience is positively related to teacher self-efficacy [27], job commitment, and meaningful work [2,28], the perception of well-being [26,29], and the subjective perception of health [30]. A lack of resilience is negatively associated with stress, exhaustion, and difficulties maintaining discipline in the classroom [8,31,32,33]. In addition, there is evidence of a mediating role between abuse (physical and emotional) in childhood and satisfaction with life in teachers [34].

Conversely, resilience in teachers is positively related to the development of resilience in children, being an important lesson during the school years, particularly for those students from vulnerable contexts or who are having difficulties in different areas [13]. Schools provide a suitable setting to promote resilience by building safe spaces where children can form a positive self-image through their teachers’ expectations and knowledge, helping them feel accepted and valued [13]. As Salvo-Garrido et al. [35] indicate, a positive teacher–student relationship is relevant to developing socio-educational resilience processes, reflected in different aspects such as student academic performance. In this respect, students’ integral development is considered to be fundamental, i.e., the commitment of teachers to the personal, social, and emotional development of children so that they can adapt to different situations in the present and future, and not just to a focus on acquiring academic knowledge. Therefore, the school should promote resilience [36,37,38]. Thus, teachers can be role models as they are often significant figures, being among the primary caregivers and sharing a significant amount of time with each child. They can therefore become a resilience tutor for their students, reinforcing the development of greater autonomy in decision making, improved problem-solving skills, and increased self-confidence [37].

Grotberg [39] refers to there being different sources of resilience. The first is the development of inner strengths (“I Am”), which corresponds to the perception of oneself as a likable, self-respecting, and responsible person who experiences positive emotions when helping others. The second is external supports (“I Have”), which are connected to the close social circle, i.e., those with whom there is a bond of affection and trust, who teach and foster autonomy, and who are there when needed. The third is problem-solving skills (“I Can”), which reflect social skills (expression of emotions) and the ability to resolve difficult situations. 

In the same vein, Saavedra and Villalta [15,40] suggest that a resilient response is the expression of the view of a problem related to how complex situations are resolved in the main; the view of oneself that integrates the cognitive and affective characteristics mobilized to deal with problems, in addition to the person’s system of beliefs (basic conditions) and the social bonds that are derived from life experiences. Resilience is constructed through the person’s history; therefore, it is constituted in early bonds that are learned. However, it is in a constant process of change, reflecting how a person interprets the situations in which they live and how they act when faced with problems [15]. In this way, the proposal by Saavedra [40] and Saavedra and Villalta [15] is complemented by what is pointed out by Grotberg [39]. 

Saavedra and Villalta [15] developed a scale to assess resilience using a sample that included people aged between 15 and 65. The instrument has 12 correlated dimensions, demonstrating adequate psychometric properties of reliability and validity for a sample of Chilean adolescents and adults. The dimensions are related to the four areas of depth previously described, from resilient behavior to the person’s system of beliefs (basic conditions, view of oneself, view of the problem, resilient response), as well as to the sources of resilience (I have, I am, I can) proposed by Grotberg [39]. The 12 dimensions are identity, autonomy, satisfaction, pragmatism, links, networks, models, goals, affectivity, self-efficacy, learning, and generativity (Table 1).

Although there is some psychometric evidence for the scale, some observations must be made to optimize its instrumental analysis, given that it is useful for various settings. First, the initial article [15] presented an interesting proposal, but its strength was conceptual and not empirical, since no psychometric evidence was indicated after the development of the items. However, subsequent studies have covered that knowledge gap. 

Villalta-Paucar et al. [41] sought to obtain a brief version for Peruvian and Chilean adolescents and began with an individual item analysis. For this procedure, the authors mention that “those that did not present values between −1.5 and +1.5 were eliminated” (p. 4); however, they do not indicate the statistic or parameter to which they refer, which could confuse the reader, given that there are more precise criteria to obtain brief versions such as factor loadings or potential unidimensionality [42]. With respect to the study by Moscoso-Escalante and Castañeda-Chang [43] conducted on Peruvian older adults: although it began appropriately with an approach based on a factor analysis (principal axis) and Promax rotation, a principal component analysis (PCA) was subsequently used, which does not distinguish between common and specific variance in the items, which artificially inflates the factor loadings and is therefore not recommended for psychometric studies [44]. Additionally, some complex items were not reported, i.e., those that received significant influence from more than one factor (e.g., item 23), making the interpretation of the scores difficult. The interfactor correlations were not reported either. 

In addition, two instrumental studies developed in Mexico were found. The first [45] supported the structural analysis in the PCA. Although this analysis was executed at three points (“I am”, “I have”, and “I can”), it is understood that this is a valid resource considering that the sample size is modest and the group is difficult to access (oncology patients); however, in addition to the intrinsic limitation of the PCA, it is worth noting that the secondary factor loadings were not shown and the difference between the variance explained by the first and second factors is very wide in the three analyses, which could suggest a latent unidimensionality for each of the blocks examined. The second study focused on healthcare personnel [46] and presented some criteria that have now been surpassed. For example, the initial filter consisted of the corrected item–total correlation, but this assumes the unidimensionality of the scale without previous evidence. However, the other criteria based on the number of items per factor and factor loadings of low magnitude are more acceptable. 

In terms of the applications of the scale, Flores González et al. [47] studied the resilience level and its association with stress, anxiety, and depression in formal caretakers of institutionalized older adults in long-term-stay facilities during the COVID-19 pandemic. It was observed that the greater the resilience, the better they coped with anxiety and stress. On the other hand, Saavedra and Cifuentes [48] refer to secondary school students obtaining an average score for the resilience variable; specifically, they performed better in the dimensions of models, generativity, and self-efficacy, with no significant differences being observed between men and women. Coincidentally, Fuentes and Saavedra [49] also reported that adolescents aged between 15 and 19 in high school showed a mean level of resilience, with higher scores in the dimensions of models, generativity, and learning. By contrast, they had lower scores in identity, affectivity, and links. Villalta and Saavedra [50] posited that students from vulnerable settings must have the ability to overcome difficult situations, for which it is necessary to rely on their immediate environment, seek solutions to problems, and value the lessons that both negative and positive experiences can teach them; hence, students’ experiences at school are related to resilience. 

It is important to note that there are other scales widely used to assess this construct in adults, such as the Connor–Davidson Resilience Scale [51]. The Brief Resilience Scale (BRS) [52], which was validated and adapted for Chilean university students, consists of six items and presents adequate psychometric properties [53]. However, it is important to bear in mind that neither the CD-RISC nor the BRS was originally constructed in the Latin American context [54,55]. They both focus on two groups (adolescents and university students) that differ from teachers in terms of evolutionary and social issues, which means that the structure cannot be extrapolated to teachers; moreover, they focus mainly on personal resources, paying less attention to social and community aspects [55,56]. This approach contrasts with more recent research that conceives of resilience as a dynamic process that integrates personal, social, and contextual resources [9,13,14]. 

In this respect, the SV-RES60 developed in Chile [15] complements personal resources with those from social networks, which are available to provide help and support in solving problems, thus promoting a positive interaction that influences the person’s resilient behavior. It is worth noting that resilience focused exclusively on individual characteristics can generate negative emotions when people face difficulties, while reducing the responsibility of other agents such as institutions and individuals [57,58], and it is in this context that the SV-RES60 addresses this limitation.

Teachers’ resilience is a relevant construct that has attracted researchers’ interest due to its importance for teachers, students, and schools [1,12,46]; however, the empirical exploration regarding resilience is still considered to be insufficient [59]. In this sense, individual factors and, to a greater extent, contextual factors, can be subjected to interventions through the use of school programs relevant to fostering safe and collaborative environments beneficial to all members of the school community [60]. The support that can be offered to teachers is significant with regard to increasing resilience [12,61]; therefore, the interventions implemented can contribute towards increasing the quality of education and empowerment against adversity [62], understanding that strengthening teachers’ skills systemically affects the entire school community [38]. Thus, knowing teachers’ needs is crucial to generating appropriate and effective interventions that contribute to well-being and quality education [32]. For this, measuring instruments that are valid and reliable for a particular population, such as Chilean teachers who work in elementary education, are required. 

Considering the abovementioned issues, i.e., the theoretical and empirical relevance of the resilience construct, the following hypothesis is proposed: the scores on the SV-RES60 Scale will present a factor structure of 12 correlated factors—as offered in the original proposal [15]—and adequate levels of reliability for the Chilean context. Therefore, this study endeavored to analyze the psychometric properties of the SV-RES60 Scale in a sample of Chilean elementary school teachers.

## 2. Materials and Methods

### 2.1. Design

The investigation is based on an instrumental design, where the psychometric properties of the SV-RES60 Resilience Scale for adults were studied [63]. 

### 2.2. Participants

The study population comprises all the teachers who work in primary education (1st to 8th grade elementary) in public schools in Chile, N = 85,298. A stratified random sample was selected considering the following strata: region, habitat (urban, rural), type of funding (public and subsidized schools), and gender. Stratified, multistage probability sampling was used, with a reliability of 95%, a sampling error of 2.5%, and a variance p = q = 0.5 [64]. The expected sample was 1576 teachers, representing 1.85% of the population. Finally, a sample of 1406 first-cycle teachers (22.5% men; 77.5% women) was obtained, with an average age of 41.4 years (SD = 10.8 years). A total of 61.5% of the teachers were at most 42 years old, and 1.2% were over 65. The number of years of teaching experience ranged from less than one year to 48 years, with an average of 14.2 years (SD = 10.1). A total of 250 schools participated in the study: 203 were in urban areas and 47 in rural areas, distributed across the country, and 83.6% of the schools were public and 16.4% were subsidized. In Chile, such schools typically belong to similar socioeconomic contexts: low and middle socioeconomic strata. Sixty-five percent of the teachers worked in public schools. The ages of the children in primary education range from 6 years old (first grade), to 13 years old (eighth grade). These ages are averages and are based on the ages that students would be at the beginning of the school year if they entered the education system at the typical age. 

### 2.3. Instrument

Sociodemographic questionnaire: this instrument was developed by the researchers to collect relevant data on the study participants and consisted of closed-ended questions regarding age, gender, region, commune, name of school and type of school (public, subsidized), residence (urban, rural), professional title, years of experience, and belonging to an ethnic group.

The SV-RES60 Resilience Scale for young people and adults was created and validated in Chile by Saavedra and Villalta [15] for an urban population between 15 and 65 years. This instrument consists of 60 items measured on a Likert-type scale with five categories (1 = Strongly disagree, 5 = Strongly agree) that account for a general level of resilience and 12 factors: Identity (I am/basic condition), Autonomy (I am/view of oneself), Satisfaction (I am/view of the problem), Pragmatism (I am/resilient response), Links (I have/basic condition), Networks (I have/view of oneself), Models (I have/view of the problem), Goals (I have/response), Affectivity (I can/basic condition), Self-efficacy (I can/view of oneself), Learning (I can/view of the problem), and Generativity (I can/response), with an administration time of 20 min. An adequate reliability level is reported, with a Cronbach’s alpha of 0.96, and appropriate validity, with a Pearson’s linear correlation coefficient of 0.76.

### 2.4. Procedure

In the first stage, all public-school principals, mayors, and directors of local education services were contacted by email, because public schools in Chile are under the administration of municipalities, and local education services are administered by the Chilean Ministry of Education. In the second stage, visits were made to the schools, requesting meetings with the principals and regional education authorities, presenting the project, and inviting them to participate. In addition, some regional authorities held meetings with all the school principals in their jurisdiction to present the study to them and give them lectures on the topic and encourage them to participate. All those who accepted the invitation were sent information about the study, a link containing the informed consent form, sociodemographic questionnaire, and the scales. The principals informed and invited the teachers to participate voluntarily, deciding the day and time to do so. Once they entered the link, the teachers were required to read the informed consent form and accept the invitation to continue answering; otherwise, the process was ended.

The data were collected through a computerized platform (Question Pro) containing the SV-RES60 Scale. In addition, there was a questionnaire with sociodemographic questions, and the informed consent form, which explained, in order to protect the ethical principles of the project, the aim of the study, the voluntary nature of the study, and the risks and benefits, etc. Visits were scheduled, and application of the study took place in the schools to ensure the sample size. The study has the approval of the Scientific Ethics Committee of the Universidad de La Frontera, Chile (Evaluation File N° 053_21; Study Protocol Page N° 019/21).

### 2.5. Analytical Approach

Preliminarily, the approximation to univariate normality was analyzed with the items on the SV-RES60 Scale, considering the magnitude of the skewness and kurtosis of each item (between −1 and 1), as well as multivariate normality with Mardia’s kurtosis coefficient (G2) [65], where magnitudes below 70 would indicate no significant departure from normality [66].

To evaluate the factor structure of the SV-RES60 Scale, six models were evaluated with exploratory structural equation modeling (ESEM) [67]. These models are: a unidimensional one (M1), which consists of one factor that influences the 60 items; a model of three oblique factors (M2): F2.1, F2.2, F2.3; another also of three oblique factors (M3) but with a different configuration: F3.1 (Items 1–20, 23, 24), F3.2 (Items 21, 22, 27–36), and F3.3 (Items 37–60); a model of four oblique factors (M4): F4.1, F4.2, F4.3, and F4.4; a model of 12 oblique factors (M5). Finally, a bifactor model that included a general factor (GF) and 12 specific factors was also used.

In terms of the estimation method, the weighted least squares means and variance adjusted method was used (WLSMV) [68], which recommended for analyzing ordinal variables [69] in a broad range of sample sizes [70]. In addition, WLSMV makes no distributional assumptions about the observed variables [71]. Consequently, the robust standard errors of the structural coefficients are more precise than those obtained with MLR (maximum likelihood robust) and ULSMV (unweighted least squares with mean and variance adjustment) in all skewed data situations [72]. Mplus v. 8.4 software was used [73].

The option of using ESEM analysis was based on the following reasons: first, ESEM is considered a fundamentally confirmatory technique [74], but it is more flexible and has fewer errors of identification and specification than confirmatory factor analysis (CFA) [74,75]. This means that some factor parameters (e.g., secondary loadings) not modeled in the CFA are modeled in the ESEM, allowing for a more precise assessment of the model. Second, it provides a more suitable representation of the data in terms of fit, especially for confirmatory purposes [74,75]. Third, it achieves more exact estimations of the relationships among latent factors [76,77], since, in the CFA, the interfactor correlations tend to be overestimated due to the absence of a cross-loading model. Thus, if the ESEM fits better to the data than the CFA, it is likely that the estimation of the factorial correlation will be substantially less biased in the ESEM model than in the CFA model [74]. Fourth, ESEM models tend to be more closely aligned with the theoretical representation of the construct that the instrument is intended to measure [78], since, in the real world, psychological constructs are complex, and it is unlikely that the factor independence proposed by the CFA will appear (or that a factor influences only one item). 

In the case of the estimations of the ESEM models, target rotation was used, which makes it possible to use this technique in a confirmatory way since it produces the rotated solution closest to a prespecified load configuration [67]. It provides a more robust model a priori and facilitates the interpretation of the results [74].

Several assessment criteria were used to evaluate the oblique ESEM models. The first consisted of the magnitude of the most frequent goodness-of-fit indices: WLSMV- χ^2^, comparative fit index (CFI > 0.90) [79], Tucker–Lewis index (TLI > 0.90) [79], root mean square error of approximation (RMSEA < 0.08) [80] and standardized root mean square residual (SRMR < 0.08) [81]. As the second criterion, factor loadings greater than 0.50 in theoretical factor were considered acceptable [82], and in addition, the simplicity of the item was evaluated using the index of factorial simplicity (IFS) [83], where values above 0.70 indicate that the item is predominantly influenced by only one factor [83,84].

In the case of bifactor modeling, additional criteria were used to evaluate the strength of the GF. The unidimensionality of the scale was evaluated through the explained common variance (ECV), percentage of uncontaminated correlations (PUC), and percentage of reliable variance (PRV) [85,86]. In that sense, ECV and PUC values over 0.70 indicate slight relative bias, and the common variation can be considered as essentially unidimensional [87]. Regarding the PRV, values over 0.75 indicate strong evidence for the use of subscales [71]. To evaluate the reliability of the scale, Omega hierarchical (ωh) and Omega hierarchical by subscale (ωhs) coefficients were used [87]. These indicators were obtained using a Microsoft Excel-based tool [88]. Finally, we used McDonald’s Omega (ω), the H coefficient [89], and Cronbach’s alpha to evaluate reliability.

Subsequently, measurement invariance was analyzed according to sex and type of school (public and subsidized) with a multi-group factor analysis using the unidimensional model. An assessment was made of the configural invariance (equivalence of the internal structure), weak invariance (equivalence of the factor loadings), and strong invariance (equivalence of the thresholds) [90]. The degree of invariance was evaluated by considering the variation in the CFI and RMSEA indices, and an acceptable degree of invariance was determined if ΔCFI > −0.01 and ΔRMSEA ≤ 0.015 [91].

## 3. Results

The 60 items presented mean values varying between 3.99 (item 41) and 4.65 (item 11); standard deviations varying between 0.67 (item 50) and 1.05 (item 26). Skewness was negative in every case and less than −1 (−2.81, item 11; −1.01, item 41), and kurtosis values varied between 0.61 (item 41) and 9.59 (item 50). Mardia’s coefficient, G2 = 6022.41 (*p* < 0.05). These results show that the items present multivariate skewness and kurtosis, and multivariate asymmetry, skewness, and kurtosis.

Table 2 shows the results of the six adjusted models. Models M4, M5, and M6 presented fit indices of acceptable magnitude, although the outlook is different at the item level. As for the M4 model, only 43.3% of the estimated standardized loadings (26 items) exceeded the value of 0.5 in the theoretical factors, and 50% of the items demonstrated factor complexity; therefore, M4 was not considered to be viable. M5 presented similar behavior, where 25 items (41.7%) presented factor loadings above 0.50 in their theoretical factors, and 65% of the items showed factor complexity; therefore, M5 was not considered to be a model that reflected the multidimensionality of resilience.

As for the bifactor model, the primary loadings of the specific factors were generally very low and mostly nonsignificant (Table 3 and Table 4). In addition, the evidence suggested that the GF had more support than the specific factors in view of the magnitude of the ECV (>0.80), PUC (>0.90), and ωh (>0.95), whereas the ωhs were low (<0.30) (Table 3 and Table 4). Finally, the PRV for the general factor was 98.4%, indicating strong evidence in favor of using the total score. By contrast, all the PRV values for the subscales were below the 75% limit, which contradicted the use of scores by dimension.

With respect to reliability, Cronbach’s alpha coefficient and the Omega coefficient presented high reliability values: 0.981 and 0.991, respectively. The H index of the general factor was 0.988, which signified a high correlation between the general factor and the optimally weighted items.

Regarding measurement invariance, the results showed that an acceptable weak invariance was achieved, although strong invariance was not (Table 5). This indicated that although the factor structure of the scale is similar between men and women, as well as between teachers from different types of schools, the group scores cannot be compared directly.

## 4. Discussion

Given the role they play, the resilience of teachers is fundamental, particularly in the initial formative stages, having implications not only in the teaching of content, but more broadly, in the cognitive, social, and emotional development of their students [1,35,37]. On the other hand, the current setting is distinguished by change, and new difficulties arise that require teachers to adapt in order to accomplish their proposed objectives; this is in addition to the demands of the school itself, including the high workload. For these reasons, it is challenging work [2,6,9,10]. In this light, it is relevant to understand teachers’ perceptions about their own resilience in order to support them and design appropriate interventions in an educational context [32].

This study sought to analyze the psychometric properties of the SV-RES60 Resilience Scale in a sample of Chilean elementary school teachers. This scale was developed by Saavedra and Villalta [15] and is supported theoretically by a proposal by Grotberg [39], who posits different sources of resilience (I am, I have, I can). Similarly, Saavedra and Villalta [15,40] state that resilience manifests in the resilient response that is derived from the view of the problem, the view of oneself, and the basic conditions that relate to the person’s belief system; therefore, it is built over time, affording the self a sense of continuity while it is transformed in its interactions with the environment. In this study, six factor models were tested; however, the oblique models (models four and five) and bifactor model (model six) showed an adequate fit. In the case of the oblique models, a significant percentage of the factor loadings did not fulfill the established standards (<0.50), in addition to presenting factor complexity that affected the interpretation of the factors. Thus, the results supported the proposal of a bifactor ESEM model in which resilience was represented by a general latent factor, i.e., where the sources of resilience and the resilient response converge and manifest simultaneously. Consequently, the initially raised hypothesis indicating that the scores from the SV-RES60 Scale would present a factor structure of 12 correlated factors, as referred to in Saavedra and Villalta [15], was rejected.

These results differ from the structures proposed in previously conducted psychometric studies because the results are not directly comparable when using analyses not recommended for the study of an internal structure [41,45]. In addition, unidimensionality was not addressed as an alternative, even in the presence of indications such as a significantly higher explained variance in the first factor [45]. Notably, none of the previous studies analyzed secondary loadings or factor complexity, which is crucial to understanding the internal structure in conceptually complex constructs. In addition, this study represents a methodological advance by incorporating exploratory structural equation modeling (ESEM) for the analysis of the SV-RES60 Scale, and presents characteristics that underscore its usefulness, given that this construct tends to have unidimensionality-oriented behavior [92,93]. Finally, the robust reliability of the construct and scores will make its use in applied settings possible since it substantially minimizes measurement errors.

Regarding the invariance analysis performed according to sex and type of school, it should be noted that, since only weak invariance was fulfilled, it is recommended that the scores of each group be interpreted independently, given that the comparisons could be biased due to personal, contextual, or cultural factors that could be clarified in future studies. It should be noted that this finding does not detract from the usefulness of the instrument for mass evaluations, but it does limit its use in comparative studies.

It is essential to highlight that this study provides a valid and reliable instrument to assess resilience in the elementary school teacher population. This could enable intervention programs to be executed based on the outcomes of the resilience measurement taken before implementing the intervention, thus making it more relevant and compatible with teachers’ needs. At the same time, this scale could be used to perform the final evaluation of this process to determine whether people have experienced changes. This would help teachers to feel supported, especially those with lower levels of resilience, by the institution, their colleagues, and other professionals who form part of the interdisciplinary team, allowing them to express the difficulties and strengths they perceive at a personal and contextual level, while developing the skills to cope with difficult situations [62]. Therefore, this study contributes to decision making in schools, benefiting not only teachers, but the entire educational community systemically [38]. This becomes relevant if it is considered that resilience is related to teachers’ self-efficacy [27], job commitment, perception of well-being, and health [26,29,30], and that a lack of resilience is associated with stress and burnout [8,32,33]. Therefore, the scale’s straightforward approach will have favorable effects on teachers and their environments.

This instrument will contribute to the assessment of resilience in elementary school teachers, supporting decision making in schools, and this will not only benefit teachers, as it is through teachers that entire school communities are systemically affected [38]. It must be noted that resilience is related to teacher self-efficacy [27], job commitment, and the perception of well-being and health [26,29,30], and a lack of resilience is associated with stress and burnout [8,32,33]. Consequently, the actions taken to support teachers by increasing their resilience would also enhance the quality of teaching and the empowerment of teachers in the challenging situations they face with their students, classes, and schools [62]. 

Concerning the limitations of the study, it can be pointed out that, in this study, elementary school teachers working in private schools did not participate. Furthermore, this is a cross-sectional study providing limited evidence in terms of temporality, and this is a self-report instrument that reflects the participants’ perceptions. It is recommended that an analysis be performed using another analytical framework (e.g., item response theory) to determine which items show differential functioning due to the absence of strong invariance. Similarly, it is recommended that future lines of enquiry provide psychometric evidence of a briefer SV-RES60 Scale, considering the relevance of having instruments with adequate validity and reliability that assess resilience, but with fewer items, thereby improving its application through the shorter amount time required by participants to complete the questionnaire [92]. In addition, psychometric studies could be conducted to provide evidence of validity related to other variables, convergent or divergent. Moreover, cross-cultural research could be conducted to evaluate the validity and reliability of this instrument in different populations, with the possibility of comparing the results with those of other instruments. Finally, correlations between teacher and student resilience and other constructs, such as stress and perceived self-efficacy, could be developed, because the resilience of teachers could affect that of their students.

## Figures and Tables

**Table 1 behavsci-13-00781-t001:** Description of the 12 factors of resilience presented by Saavedra and Villalta [15].

Factor	Definition
Identity: F1 I am-Basic Condition.	Judgments that reflect the way a person interprets actions and circumstances in the social and cultural context. This defines the person in a reasonably stable way, giving them a sense of historical continuity.
Autonomy: F2 I am-View of oneself.	Judgments that reflect the person’s contribution to their sociocultural surroundings.
Satisfaction: F3 I am-View of the problem.	Judgments that show the interpretation a person makes about a problematic situation.
Pragmatism: F4 I am-Resilient response.	Judgments that give an account of how a person interprets their actions.
Links: F5 I have-Basic conditions.	Judgments about the value of primary socialization considering the affective and social bonds a person has that reflect their personal history.
Networks: F6 I am-View of the problem.	Judgments related to a person’s affective bonds with their close surroundings.
Models: F7 I have-View of the problem.	Judgments that reflect the person’s beliefs about the support their close social circle can offer in problematic situations.
Goals: F8 I have-Responses.	Judgments about the value of the proposed goals include people close to them who are considered essential and available to support them in difficult times.
Affectivity: F9 I can-Basic conditions.	Judgments about the possibilities of managing emotions and expressing them while building trusting relationships.
Self-efficacy: F10 I can-View of oneself.	Judgments that reflect the evaluation of the possibilities of successfully solving the problem.
Learning: F11 I can-view of the problem.	Judgments indicating the problematic situation can be understood as a learning opportunity.
Generativity: F12 I can-Response	Judgments the person makes that indicate the possibility of asking for help to solve a problematic situation.

**Table 2 behavsci-13-00781-t002:** Goodness-of-fit of models M1 to M6 for the SV-RES60 Scale.

Model	X^2^	df	CFI	TLI	RMSEA	90%CI RMSEA	SRMR
M1	14581.7	1710	0.886	0.882	0.073	[0.072, 0.074]	0.097
M2	12754.4	1593	0.901	0.890	0.071	[0.069, 0.072]	0.038
M3	12247.9	1482	0.902	0.891	0.072	[0.071, 0.073]	0.038
M4	8504.8	1536	0.938	0.929	0.057	[0.056, 0.058]	0.027
M5	2854.7	1116	0.985	0.976	0.033	[0.032, 0.035]	0.013
M6	599.2	1068	0.986	0.977	0.032	[0.030, 0.033]	0.012

**Table 3 behavsci-13-00781-t003:** Standardized factor loadings resulting from the bifactor ESEM model and indicators of unidimensionality and reliability for the SV-RES60 Resilience Scale of the GF and factors F1 to F6.

Item	GF	F1	F2	F3	F4	F5	F6
1	0.710	**0.409**	−0.050	−0.053	0.098	−0.019	−0.112
2	0.689	**0.141**	0.435	−0.064	−0.015	−0.014	−0.109
3	0.727	**0.377**	0.112	−0.073	−0.004	−0.024	−0.107
4	0.692	**0.351**	0.179	0.072	0.077	0.024	0.07
5	0.715	**0.429**	0.080	0.045	0.095	−0.022	0.114
6	0.605	0.150	**−0.016**	0.215	0.065	−0.119	0.251
7	0.691	0.112	**0.043**	0.267	0.033	−0.097	0.152
8	0.695	0.037	**0.624**	−0.013	0.025	−0.008	−0.023
9	0.739	0.168	**0.429**	0.119	0.004	−0.033	−0.034
10	0.683	0.196	**0.125**	0.155	0.071	0.076	0.127
11	0.749	0.219	0.001	**0.147**	0.166	−0.034	−0.075
12	0.672	0.113	0.025	**0.285**	0.125	−0.089	−0.058
13	0.680	−0.034	0.034	**0.482**	0.111	−0.036	−0.012
14	0.707	−0.019	0.107	**0.387**	0.083	0.009	−0.05
15	0.696	0.018	0.087	**0.356**	0.134	0.302	−0.117
16	0.648	−0.033	0.106	0.17	**0.304**	−0.028	−0.068
17	0.782	0.109	0.011	0.24	**0.222**	−0.064	−0.033
18	0.758	−0.031	0.048	0.016	**0.498**	−0.026	−0.021
19	0.532	0.091	−0.050	0.052	**0.471**	−0.093	0.09
20	0.760	0.059	−0.037	0.008	**0.517**	0.006	−0.003
21	0.743	−0.001	−0.034	0.044	−0.05	**0.265**	0.125
22	0.647	0.039	−0.002	0.02	−0.02	**0.516**	0.192
23	0.752	−0.027	−0.001	0.116	−0.019	**0.576**	0.025
24	0.818	0.003	0.017	−0.228	0.047	**0.164**	−0.062
25	0.883	0.03	−0.057	−0.098	−0.058	**0.174**	0.01
26	0.613	−0.016	−0.074	−0.128	0.06	0.049	**0.211**
27	0.807	−0.024	−0.058	−0.029	−0.039	0.093	**0.361**
28	0.788	−0.03	−0.012	−0.067	−0.027	0.095	**0.389**
29	0.825	0.01	−0.003	−0.004	−0.07	0.038	**0.200**
30	0.844	−0.001	0.004	0.031	−0.067	0.11	**−0.079**
31	0.804	−0.012	−0.109	0.038	−0.093	−0.062	0.077
32	0.797	−0.033	−0.065	0.027	−0.105	−0.081	0.056
33	0.804	−0.078	0.024	0.036	−0.081	0.076	0.068
34	0.764	0.055	−0.058	−0.017	0.029	0.052	−0.028
35	0.798	−0.025	0.000	−0.022	0.012	0.022	0.117
36	0.796	0.012	0.031	−0.117	0.109	−0.005	−0.046
37	0.831	0.016	0.123	−0.049	−0.006	0.016	−0.08
38	0.804	−0.049	0.019	−0.015	0.046	0.094	0.122
39	0.849	−0.042	−0.008	−0.004	0.000	0.011	0.004
40	0.811	−0.063	−0.054	−0.100	0.164	0.074	0.003
41	0.609	0.005	0.045	−0.027	0.045	−0.076	0.142
42	0.648	0.079	−0.014	−0.005	−0.093	0.04	0.063
43	0.620	−0.027	−0.005	0.047	−0.132	0.02	−0.021
44	0.826	0.019	−0.077	−0.144	0.017	0.015	−0.135
45	0.804	0.008	−0.098	0.057	−0.094	0.033	−0.178
46	0.824	−0.149	−0.047	0.000	0.068	−0.021	−0.203
47	0.702	−0.203	0.107	−0.029	0.024	−0.144	0.053
48	0.733	−0.089	0.010	−0.038	0.074	−0.093	0.074
49	0.743	0.091	−0.111	−0.033	0.002	−0.08	−0.03
50	0.774	−0.019	0.000	−0.052	−0.044	−0.005	0.027
51	0.678	−0.016	−0.009	0.021	−0.028	−0.061	0.001
52	0.760	−0.121	0.079	0.085	−0.094	−0.021	−0.025
53	0.781	0.009	−0.033	0.036	−0.011	0.039	−0.034
54	0.737	0.107	−0.099	0.056	−0.042	0.003	−0.065
55	0.787	−0.137	0.140	−0.077	0.055	−0.027	−0.002
56	0.810	−0.052	0.015	−0.094	0.139	0.039	−0.058
57	0.797	0.016	0.016	−0.046	0.037	−0.049	−0.057
58	0.820	0.084	−0.051	−0.034	−0.069	−0.018	−0.038
59	0.715	−0.088	0.009	−0.017	−0.013	−0.109	0.026
60	0.846	−0.039	−0.040	0.026	−0.064	−0.068	0.038
ECV	0.812	0.015	0.014	0.015	0.021	0.018	0.009
ωh	0.975						
ωhs		0.169	0.096	0.162	0.227	0.152	0.066
H	0.988	0.430	0.468	0.423	0.531	0.498	0.297
PUC	0.932						
PRV	0.984	0.189	0.111	0.183	0.251	0.163	0.072

ECV: Explained Common Variance; ω: McDonald’s Omega; ωh: Omega Hierarchical; ωhs: Omega Hierarchical Subscale; H: Construct replicability; PUC: Percentage of Uncontaminated Correlations; PRV: Percentage of Reliable Variance.

**Table 4 behavsci-13-00781-t004:** Standardized factor loadings resulting from the bifactor ESEM model and indicators of unidimensionality and reliability for the SV-RES60 Resilience Scale of the GF and factors from F7 to F12.

Item	FG	F7	F8	F9	F10	F11	F12
1	0.710	−0.052	−0.095	0.076	−0.176	−0.026	−0.027
2	0.689	−0.048	−0.145	0.016	−0.095	−0.066	−0.071
3	0.727	−0.127	−0.071	0.054	−0.152	−0.113	0.041
4	0.692	0.020	0.006	−0.02	0.198	0.066	−0.026
5	0.715	0.024	0.048	−0.025	0.09	0.064	0.050
6	0.605	0.097	−0.151	0.052	−0.002	−0.29	0.104
7	0.691	0.110	−0.216	0.016	0.037	−0.254	0.017
8	0.695	−0.040	−0.017	−0.024	−0.006	0.013	−0.047
9	0.739	−0.039	0.158	−0.039	−0.006	0.000	0.058
10	0.683	−0.056	−0.021	0.022	−0.088	−0.01	0.044
11	0.749	−0.037	−0.016	−0.140	0.053	0.085	−0.106
12	0.672	−0.021	−0.064	−0.012	−0.105	0.211	−0.161
13	0.680	−0.045	−0.107	0.005	−0.113	0.144	−0.126
14	0.707	0.143	−0.103	0.035	0.042	−0.106	−0.096
15	0.696	0.006	−0.041	0.068	−0.029	−0.116	0.004
16	0.648	0.002	−0.070	−0.130	0.019	0.056	−0.007
17	0.782	−0.112	0.211	−0.085	0.049	0.004	0.002
18	0.758	−0.069	0.042	−0.033	0.002	−0.045	−0.009
19	0.532	−0.026	0.036	0.006	0.013	0.018	−0.067
20	0.760	−0.058	0.004	−0.054	−0.033	−0.018	0.039
21	0.743	0.073	0.004	0.013	0.068	−0.1	0.001
22	0.647	−0.054	0.007	0.013	−0.035	0.105	0.032
23	0.752	−0.008	0.010	0.036	−0.034	−0.008	0.024
24	0.818	−0.115	−0.017	−0.088	0.034	−0.044	−0.178
.25	0.883	−0.156	0.041	−0.099	−0.038	0.009	−0.166
26	0.613	−0.054	0.033	0.041	−0.072	0.021	−0.107
27	0.807	0.101	−0.017	−0.027	0.009	−0.027	−0.069
28	0.788	0.140	−0.002	0.009	−0.011	−0.025	−0.026
29	0.825	0.193	0.001	−0.011	−0.116	0.009	−0.008
30	0.844	0.064	0.146	−0.029	−0.196	0.089	−0.044
31	0.804	**0.360**	0.008	−0.061	−0.148	0.016	−0.049
32	0.797	**0.414**	−0.001	−0.063	−0.111	0.004	−0.038
33	0.804	**0.394**	0.042	−0.049	0.065	−0.061	−0.025
34	0.764	**0.439**	0.151	0.047	0.191	0.053	−0.181
35	0.798	**0.457**	0.062	0.03	0.025	−0.019	−0.037
36	0.796	0.225	**0.351**	0.054	−0.08	−0.039	0.055
37	0.831	0.120	**0.349**	−0.031	−0.058	0.059	0.005
38	0.804	0.301	**0.140**	0.065	−0.069	−0.068	0.068
39	0.849	−0.002	**0.444**	−0.038	−0.001	−0.066	0.046
40	0.811	0.103	**0.231**	−0.02	−0.039	−0.066	0.05
41	0.609	−0.109	0.056	**0.676**	−0.118	0.104	−0.08
42	0.648	−0.081	0.019	**0.575**	0.080	0.090	−0.061
43	0.620	0.004	−0.036	**0.469**	0.058	−0.127	0.017
44	0.826	−0.071	−0.073	**0.097**	0.138	−0.055	0.110
45	0.804	0.055	−0.028	**0.356**	0.101	−0.039	0.042
46	0.824	0.017	−0.078	0.133	**0.123**	−0.012	0.094
47	0.702	−0.098	−0.024	0.161	**0.307**	0.074	−0.075
48	0.733	−0.005	−0.037	0.324	**0.019**	0.064	0.025
49	0.743	0.117	−0.038	0.184	**0.328**	0.075	0.026
50	0.774	0.007	−0.080	−0.033	**0.398**	0.161	0.096
51	0.678	−0.082	−0.080	0.017	0.143	**0.297**	0.081
52	0.760	−0.075	−0.073	0.076	0.095	**0.298**	0.078
53	0.781	0.043	−0.071	0.018	0.121	**0.285**	0.223
54	0.737	0.130	−0.101	0.113	0.099	**0.295**	0.199
55	0.787	−0.023	−0.066	−0.009	0.085	**0.162**	0.275
56	0.810	0.030	−0.117	0.022	−0.004	0.143	**0.360**
57	0.797	−0.004	0.046	−0.015	0.023	0.10	**0.411**
58	0.820	−0.039	0.038	−0.042	0.057	0.069	**0.414**
59	0.715	−0.156	0.033	−0.009	0.027	0.016	**0.342**
60	0.846	−0.219	0.160	0.047	0.025	−0.006	**0.316**
ECV	0.812	0.021	0.012	0.028	0.009	0.009	0.017
ωh	0.975						
ωhs		0.203	0.114	0.257	0.079	0.101	0.166
H	0.988	0.510	0.375	0.639	0.300	0.288	0.445
PUC	0.932						
PRV	0.984	0.213	0.121	0.277	0.088	0.113	0.176

ECV: Explained Common Variance; ω: Omega McDonald; ωh: Omega Hierarchical; ωhs: Omega Hierarchical Subscale; H: Construct replicability; PUC: Percentage of Uncontaminated Correlations; PRV: Percentage of Reliable Variance.

**Table 5 behavsci-13-00781-t005:** Factorial invariance by sex and type of school.

Model	CFI	TLI	RMSEA	90%CI	SRMR	ΔCFI	ΔRMSEA
Sex							
Configural	0.893	0.889	0.079	[0.078, 0.080]	0.099		
Weak	0.954	0.953	0.051	[0.050, 0.052]	0.102	0.061	−0.028
Strong	0.898	0.902	0.075	[0.073, 0.076]	0.099	−0.056	0.024
Type of school							
Configural	0.892	0.888	0.074	[0.073, 0.075]	0.099		
Weak	0.960	0.959	0.045	[0.043, 0.046]	0.104	0.068	−0.029
Strong	0.892	0.896	0.072	[0.070, 0.073]	0.099	−0.068	0.027

## Data Availability

The data that support the findings of this study are not available because they are confidential data.

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
