# Peer review of "Psychometric Properties of the SV-RES60 Resilience Scale in a Sample of Chilean Elementary School Teachers"

_behavsci, 2023, doi:10.3390/bs13090781_

Round 1
Reviewer 1 Report
The aim of this paper is to explore the psychometric properties of the SV-RES60 resilience scale in Chilean primary school teachers. A representative sample of teachers was used to evaluate the factorial structure of the scale in this population.
I consider this to be a relevant study since it is necessary to have instruments with psychometric properties that allow us to measure variables such as resilience in vulnerable groups, in this case primary school teachers.
I will now comment on some aspects that should be taken into account to enable the publication of the manuscript.
The title and abstract should reflect that these are primary school teachers.
The introduction correctly discusses the role of resilience in educational contexts. It discusses the benefits of teacher resilience both for the better development of the teaching role and mental wellbeing of teachers, as well as the benefit for the holistic development of students. But it would be important to include the following aspects:
- it is important to comment on other scales that have been validated in teachers in other countries, such as the Connor-Davidson scale or the Teacher Resilience Scale (ER-D) (Guerra, 2013) and justify why the SV-RES60 scale has been chosen to adapt it to Chilean teachers, and not another.
- Discuss why it is important to measure resilience in teachers, e.g. to foster these qualities, to establish teacher training programmes in resilience (or other aspects).
Regarding the Method:
- In the section on participants, the number of schools that took part in the study should be included, i.e. the number of schools to which the participating teachers belonged, this is important to understand whether the sample is representative. In addition, it should be noted how many schools were from urban, rural, etc., and whether the schools belonged to similar or different socio-economic backgrounds. This is also interesting in order to be able to analyse whether there is a difference in the level of resilience based on these variables.
It is indicated that the teachers were from the first cycle of primary education, but no comment is made as to why only this first cycle was selected and not teachers from the whole of primary education. It is strange that if the intention is to validate a test for primary school teachers, teachers from the whole of primary education are not selected. It is therefore necessary to justify this selection of teachers. Also, bearing in mind that primary education, depending on the country, may comprise children of different ages (depending on the education system), it would be necessary to explain the ages of the children in this chosen cycle who are taught by the teachers participating in the study.
It would be important to reflect the number of years of teaching experience of the participants; there are studies that indicate differences in the level of resilience of teachers according to this variable. If this was not asked, it could be included in the limitations section.
- In the instrument section, it would be very appropriate to put an example of an item in each factor, this would facilitate a better understanding of the study for those readers who are not very familiar with resilience and the instrument.
- The procedure section should be explained in more detail in order to facilitate replication and to understand how the scale was passed. It is stated that a computer platform was used, but it is also stated that the centres were visited. This is not clear: did the researchers go to the schools and pass out the questionnaires in computer rooms, or did they go to the schools to encourage teachers to participate, did they sign the informed consent at the time of passing out the questionnaire, how were teachers informed about the study and did they stay with them to pass out the scale? Please elaborate on the procedure.
It would be very interesting to discuss which socio-demographic aspects the teachers were asked about, and to include this information in the 'Instruments' section as a 'socio-demographic data sheet or form'. In addition, it could be discussed whether any of these data could improve the description of the participants, e.g. years of experience, any aspect of their basic or further training in resilience-related aspects, whether they were teachers in permanent employment or were covering a leave of absence (type of contract).
- In the section explaining the data analyses that have been carried out, it should be indicated which statistical package was used to perform the analyses.
The analyses and results support the factor structure obtained. But was any multi-group analysis carried out to confirm the gender structure of the participants? If not, it would at least be important to include in the results whether significant differences are found between men and women (including the effect size for the difference in the n of both sexes).
Although the number of participants is large, it could be commented on whether the data were tested for normality.
- Results: Since it is not possible to complement the results with convergent validity (with some stress instrument or other variable related to resilience), since as it appears in the limitations, I understand that teachers were not given another scale to control for this type of validity, it is recommended to perform an analysis of the scores obtained with the instrument, and compare the level of resilience of teachers by any socio-demographic variable studied, describing the level of resilience of teachers in general and on the basis of some variable (such as gender, age groups, years of experience, rural-urban context or other variable) and compare the results with those obtained in other studies.
- In the discussion and conclusions, the authors highlight the suitability of the scale to be applied to teachers, highlighting the factor structure found. Furthermore, the authors indicate the benefits of teacher resilience on educational processes and teacher functioning and health. But to show the importance of this work, it would be necessary to better develop what the assessment of resilience through instruments of this type brings, such as being able to establish appropriate training programmes adapted to the results obtained, to give professional support to teachers with very low levels of resilience. And even if the level of resilience were to be analysed by the 12 factors, the highest and lowest scores could be identified to indicate where training could go.
It is indicated that teacher resilience improves aspects of both teachers and students, so when proposing to conduct (in limitations) convergent and divergent validity analyses (which is essential), one could include some examples involving teachers and students, e.g. teacher stress and children's school stress, teachers' self-efficacy, children's level of resilience, ...., would be interesting future lines of research.
Finally, the references are found to be well written and quite up to date.
Author Response
Dear revisor,
Thanks for your comments, which have allowed us to improve the manuscript. Different improvements have been made to the article that are in yellow.

Reviewer 2 Report
This is a very robust manuscript. The analysis performed is very consistent and in line with contemporary recommendations for instrument validation. Likewise, the choice of scale is clearly justified and relevant. There are two aspects that should be included to improve the impact of the article and its usefulness: 1. on the one hand, an explanation becomes necessary as to why 6 different latent models have been tested for this scale (p. 5-6). This aspect is not clear and it would seem more coherent to have simply tried to replicate the original model achieved with young people and adults. 2. There is a lack of analysis of measurement invariance according to the different strata of the sample: region, habitat (urban, rural), type of funding (public and subsidized schools) and gender. At least, I think it is important to do it for gender, type of funding, and habitat. In this way, one could ensure that the instrument can be used to make comparisons between these groups and that potential differences are not due to psychometric anomalies of the instrument.
3. Given that the one-factor structure was ultimately retained, I am surprised that the authors did not attempt to reduce the length of the instrument by retaining only the more robust items. This is a common practice that would make it easier for the instrument to be used in future research. Having a unidimensional instrument with 60 items is not very practical.
4. Why are the items not included as annexes to the manuscript?
Author Response

(The authors gave the same response as above.)

Round 2
Reviewer 1 Report
Thank you for the opportunity to review the revised version of this manuscript. The authors have followed the reviewers' suggestions and this version of the manuscript is much stronger. I condsid that the revised manuscript is suitable for publication in the journal.